

**Long-term multi-source precipitation estimation with high resolution**
**(RainGRS Clim)**
**Anna Jurczyk[1], Katarzyna Ośródka[1], Jan Szturc[1], Magdalena Pasierb[1], and Agnieszka Kurcz[1]**
[1] Centre of Meteorological Modelling, Institute of Meteorology and Water Management – National Research
Institute, ul. Podleśna 61, 01-673 Warsaw, Poland
**Correspondence**: Jan Szturc (jan.szturc@imgw.pl)





**Abstract.** This paper explores the possibility of using multi-source precipitation estimates for climatological applications. A data processing algorithm (RainGRS Clim) has been developed to work on precipitation accumulations such as daily or monthly totals, which are significantly longer than operational accumulations (generally between 5 min and 1 h). The algorithm makes the most of additional opportunities, such as the possibility to complement with delayed data, access to high-quality data that are not operationally available, and the greater efficiency of the algorithms for data quality control and merging on longer accumulations. Verification of the developed algorithms was carried out on monthly accumulations through comparison with precipitation from manual rain gauges. As a result, monthly accumulations estimated by RainGRS Clim were found to be significantly more reliable than accumulations generated operationally. This improvement is particularly noticeable for the winter months, when precipitation estimation is much more difficult due to less reliable radar estimates.

## 1. Introduction

The estimation of precipitation on the ground surface with high spatial resolution is one of the most important issues in meteorology, but at the same time one of the most complex because of the very high spatial and temporal variability of precipitation, especially in the case of intense events associated with convective phenomena. This makes its precise quantitative estimation very difficult and subject to many errors. None of the available techniques, i.e. rain gauge measurements, meteorological radar measurements or satellite estimates based on measurements in different electromagnetic radiation bands, provide satisfactory precision. Consequently, different methods are being developed to combine precipitation data obtained by these techniques, with the aim of exploiting the advantages of each technique while minimising its weaknesses (Ochoa-Rodriguez et al., 2019; Jurczyk et al., 2020b; Wetchayont et al., 2023).

The generation of such multi-source precipitation estimates is currently the standard procedure used for quantitative precipitation estimation (QPE). In operational (i.e. real-time) applications, the most common time step for estimating the precipitation field is the 1-hour step, as it often follows the demand from hydrological rainfall-runoff models (Sokol et al., 2021). However, sub-hourly resolutions, such as 10-minute resolution, are also increasingly used. Such data are becoming essential, in particular as input for nowcasting precipitation forecast models, for precipitation-runoff models forecasting flash floods, which are triggered by intense but short-lived and rapidly fluctuating precipitation (e.g., Chan et al., 2016; Neuper and Ehret, 2019), or for performing analyses of the occurrence of precipitation extremes (e.g., Bonaccorso et al., 2020; Lengfeld et al., 2020; Marra et al., 2022).

However, there is also growing demand among climatologists and agrometeorologists, for example, for longer precipitation totals – of the order of days, months or years, or even entire multi-year periods – that still maintain high spatial resolution. This demand can in fact already be met, as radar observations of precipitation, providing the highest spatial resolution of all measurement techniques,



have been performed routinely for several decades. So, long series of radar as well as multi-source precipitation estimates are already available. Weather radar networks have covered a large part of the more densely populated areas of the globe, so that increasingly radar data, when supplemented with other observations, are also applied in climatological studies to provide extensive information on the multi-year variability of the precipitation field with very high spatial resolution not available with other measurement techniques (Fabry et al., 2017; Saltikoff et al., 2019a). They are also used to study the climatology of intense convective phenomena, as the high spatial resolution is particularly important in this case (Hamidi et al., 2017; Burcea et al., 2019; Voormansik et al., 2021; Hänsler and Weiler, 2022; Piscitelli et al., 2022).

Consequently, there is a need to produce reliable estimates of precipitation accumulation over longer time periods (daily, monthly, yearly, or even longer) with data from databases containing operationally generated multi-source precipitation at higher temporal resolutions, e.g. as 10-min precipitation accumulations. It turns out that simply adding up, for example, 10-min estimates does not give satisfactory results, because any quality control algorithms for precipitation observations become much more effective for longer accumulations of at least 1 hour (Morbidelli et al., 2018; Villalobos-Herrera et al., 2022). In particular, any algorithms for the adjustment of radar to rain gauge data often work too randomly when shorter accumulations are used, and the cross-checking of different types of precipitation data is then also subject to much higher uncertainty.

Generating accumulations for longer time intervals therefore provides the possibility of carrying out so-called reanalyses, i.e. re-generating the corresponding precipitation accumulation. This brings the following potential benefits: (i) data sets can be supplemented with data that were missing from the operational estimation, e.g. due to delays in their arrival at the system, (ii) in addition, data from such measurement techniques that are available too late for operational applications, or measured with a longer calculation step (e.g. daily, such as from manual rain gauges) can be used (Imhoff et al, 2021), (iii) algorithms for performing quality control on radar precipitation data and then combining them with data from other sources generally work much more effectively on longer accumulations (Wagner et al., 2012; Park et al., 2019).

Various initiatives are being undertaken to estimate precipitation data for climatological purposes with the high spatial resolution obtained from radar observations, including on a trans-national scale. One of the major initiatives in this area is the EURADCLIM (EUropean RADar CLIMatology) dataset, which is based on radar data obtained from the Operational Program on the Exchange of Weather Radar Information (OPERA) – a EUMETNET (EUropean METeorological NETwork) initiative (Saltikoff et al., 2019b), and rain gauge data obtained from the European Climate Assessment & Dataset (ECA&D) project. Both of these networks are pan-European and cover the area of most of Europe. In the EURADCLIM programme, radar quality control adapted to longer precipitation accumulation intervals, such as 1-h and daily intervals, is performed (Overeem et al., 2023). Quality control is also performed on longer rain gauge accumulations within ECA&D (Klok and Klein Tank, 2009).


The concept of generating long-term precipitation estimation presented in this paper is based on
using algorithms for quality control of the input data and combining them into multi-source estimates,
which are applied operationally to 10-min data. However, new quality control methods and new data
sources were also included – something that was not possible during the operational generation of
precipitation estimates.
Section 2 describes all input data, those available operationally as well as those used for
reanalyses. Section 3 presents the algorithm for combining precipitation data into a multi-source
precipitation field, used both operationally and for reanalyses, and Section 4 proposes a scheme for
generating long-term estimates. Section 5 shows and discusses the results of the verification of the
reanalyses of monthly totals in different seasons compared to operationally generated estimates, while
Section 6 shows an example of the system performance. Finally, Section 7 provides conclusions.

**2. Precipitation data**

*2.1. Precipitation measurement data available for the area of Poland*

Table 1 summarises the general characteristics of the precipitation data available for the area of
Poland: from in situ and remote sensing measurements, available both in real-time and after a shorter or
longer processing time, which can take up to two months (this is the case for quality control of the data
from manual rain gauges).

Table 1. In situ precipitation measurement networks available for Poland.

| Observation technique | Temporal resolution | Network density / spatial resolution | Delay |
|---|---|---|---|
| Telemetric rain gauge network | 10 min | 1 gauge per 625 km$^2$ (about 500 gauges) | 6 min (then data from more than 90% of the gauges are usually available) |
| Manual rain gauge network | 24 hrs | 1 gauge per 434 km$^2$ (about 720 gauges) | About 2 months (due to the transfer of the data and manual quality control) |
| Ground weather radar network | 10 min | About 1 km | 6 min (because the lowest scan is generated at the beginning) |
| Geostationary meteorological satellites (Meteosat and NWC-SAF software) | 5 min (in rapid scan system) | About 5-6 km | 1-5 min (due to scan strategy) |






This study uses precipitation data generated by the Institute of Meteorology and Water
Management – National Research Institute (IMGW), which performs the function of the national
meteorological and hydrological service in Poland (Szturc et al., 2018). All these data are quality
controlled by dedicated applications or systems.

*2.2. Rain gauge data*

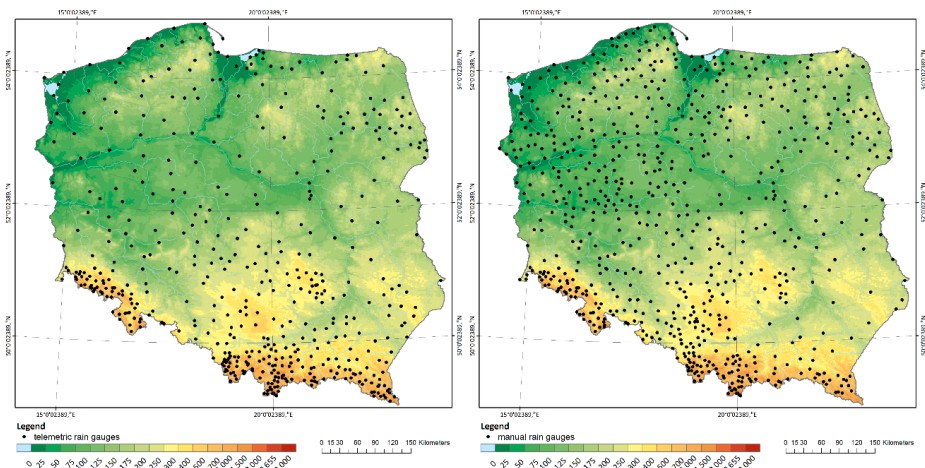



**Figure 1.** Rain gauge networks of IMGW, from left: telemetric and manual rain gauge networks.

10-min precipitation accumulations are provided operationally at IMGW by a network of
telemetric rain gauges, most of which are tipping-bucket gauges – considered one of the less accurate
of the various types of rain gauge (Hoffmann et al., 2016; Segovia-Cardozo et al., 2021) in addition to
being subject to significant failure rates. For quality control of telemetric rain gauge data, the
RainGaugeQC system is used at IMGW to perform error detection and corrections on 10-min data in
real-time (Ośródka et al., 2022).
One of the most important additional benefits of carrying out reanalyses, relative to the generation
of a real-time precipitation field, is the possibility to exploit the much more accurate measurements
performed by manual rain gauges mostly once a day. The network of such rain gauges (Hellmann type)
installed at IMGW is relatively dense, and even denser than the network of telemetric rain gauges (Fig.
1 and Table 1). These are the most accurate of the in situ point measurements, but they are available
with a very long delay of almost two months, mainly due to the human-made data quality control. In
addition, measurements from manual rain gauges are subjected to quality control in the IMGW historical
database, using standard algorithms based on procedures recommended by the WMO (WMO-No. 305,
1993, Chapter 6).




*2.3. Weather radar data*

The radar data used to generate the precipitation field estimates come from the Polish POLRAD
weather radar network, operated by the IMGW. It consists of eight Doppler radars manufactured by
Leonardo Germany (Fig. 2). They are currently being replaced by new models with dual-polarised radar
beams, and two new radars are being installed. Three-dimensional raw data, so-called volumes (raw
data), and two-dimensional products are generated by the Rainbow 5 system every 10 min (a shift to 5-
min measurement frequency is currently underway), with 0.5-km spatial resolution and a range of 250
km. For further details on the POLRAD network, see Ośródka and Szturc (2022).

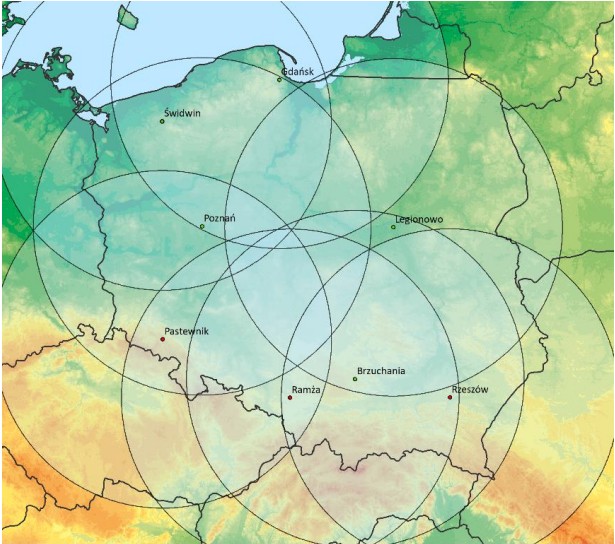


**Figure 2.** Computational domain of Poland (900 km x 800 km) with 250-km radar coverage of the weather radar
network in Poland in 2022.

The RADVOL-QC system (Ośródka et al., 2014; Ośródka and Szturc, 2022) is used to quality
control of radar data of the POLRAD network, which corrects the source 3D radar data and generates
dynamic maps of the data quality index. Merging data from individual radars into radar composite maps
is done by applying algorithms that take account of the spatial distribution of the quality index in the
radar data, which is assessed dynamically for each time step (Jurczyk et al., 2020a).

*2.4. Precipitation from meteorological satellites*



Satellite precipitation is generated by an algorithm developed at IMGW based on products

provided by the EUMETSAT NWC-SAF programme (Tapiador et al., 2019). The algorithm working
within RainGRS system is based on several NWC-SAF products, depicting the spatial distribution of
clouds and the intensity of precipitation, including convective precipitation. A detailed description of
the algorithm was presented by Jurczyk et al. (2020b).

Quality control of satellite precipitation is also carried out by the RainGRS system, taking into

account primarily which NWC-SAF products are available at a given time. The quality of satellite
precipitation, which is quantified by the quality index, is significantly lower at night-time, when visible
range-based products analysing the physical properties of hydrometeors are not available.

**3. RainGRS system**


*3.1. Merging of precipitation data into a multi-source precipitation field*


At IMGW, multi-source estimation of the precipitation field is carried out operationally by the

RainGRS system. A detailed description of this system, which combines rain gauge, radar and satellite
precipitation data summarised in Table 1, was presented by Jurczyk et al. (2020b). This combination
algorithm takes into account the quality information of the individual input data, attributed to them when
performing their quality control.

In operational work, the 10-min computational step of generating estimates of the precipitation

field is enforced by the resolution of the radar data, which is the source of the most important high-
resolution information on the spatial distribution of the precipitation field. When the radars of the
POLRAD network are replaced (process in ongoing from 2022 to 2023), all included radars will operate
with a 5-min time step. This will enable the temporal resolution of the multi-source precipitation
estimates generated by RainGRS to be increased as well.

The algorithm for combining rainfall data from different sources is based on a conditional

merging that attempts to enhance the strengths of the individual inputs and reduce the impact of their
weaknesses. It is commonly assumed that radar data is the best representation of the spatial distribution
of the precipitation field, while a network of rain gauges effectively reduces the bias of this estimation.
Satellite rainfall, in contrast, plays a mainly complementary role in the absence of other data.

First, the rain gauge values are interpolated at radar pixel resolution, employing the Ordinary

Kriging method to obtain an unbiased estimate of precipitation. The radar values at rain gauge locations
and the same method of interpolation are used to get the interpolated radar field. Subsequently, the
deviation between the measured and interpolated radar value $(R - R_{int})$ is computed and added to the rain
gauge interpolated value at each pixel of the domain, according to the following formula:
$$R_G = G_{int} + (R - R_{int}) \tag{1}$$



where $R_{int}$ is the radar precipitation interpolated from data at rain gauge locations. A satellite field $S_G$ is
obtained from an analogical formula.
It can be noted that the accuracy of the computed estimate depends on the distance to the nearest
available rain gauge, and the radar precipitation field is preferable in the case of a long distance.
Therefore, the resulting precipitation field $R_G$ is recombined with the radar precipitation field, applying
the weighted scheme, which includes the quality of individual precipitation fields to obtain a combined
*GR* field:
$$GR = \frac{R_G \cdot QI_G + R \cdot QI_R \cdot (1 - QI_G)}{QI_G + QI_R \cdot (1 - QI_G)} \qquad (2)$$
where $QI_G$ and $QI_R$ are the quality indices for gauge and radar, respectively. The quality index, *QI*, is
the dimensionless quantity ranging from 0 (for the poorest quality) to 1 (for the best data).
A combined gauge-satellite field *GS* is obtained analogically to the above procedure, where the
satellite data *S* and relevant quality field $QI_S$ are taken.
The final quantitative precipitation estimate (*GRS*) is a combination of gauge-radar and gauge-
satellite fields computed by means of the following weighted formula:
$$GRS = \frac{GR \cdot QI_d + GS \cdot (1 - QI_d) \cdot QI_S}{QI_d + QI_S \cdot (1 - QI_d)} \qquad (3)$$
where the $QI_d$ is a field of radar data quality as a function of the distance *d* to the nearest radar site.

*3.2. Generation of daily accumulations*

The basic 10-min precipitation accumulations are aggregated into different time intervals (e.g. 1-
hour, several hours, daily, or longer accumulations) depending on current needs. Due to gaps in data
that occur in operational work, sometimes these accumulations may not be complete. In order to ensure
the completeness of the accumulations, the gaps are complemented by temporal interpolation of the data
from time steps directly before and after the gap. Such averaging from neighbouring measurements is
carried out if this interval is not too long, and in the opposite case data are set to have no data value. For
example, when generating hourly accumulation, at most two consecutive 10-minute measurements are
allowed to be missing, but no more than three terms may be missing in one hour.

**4. Generation of daily and monthly precipitation reanalyses (RainGRS Clim)**

*4.1. Climatological reanalyses versus operational estimates*

Reanalysis of the precipitation fields is carried out on daily accumulations. This provides the
following benefits in terms of the reliability of the generated estimates:



1. *Complementation with data that was missing operationally due to its late arrival in the system.* For reanalyses, a time regime is not as strict as in an operational work, so data that arrived too late can be included. In the operational RainGRS, more than 90% of the rain gauge data generally arrives within six minutes, so the remaining data can be involved in reanalyses. When it comes to radar data, delays mainly affect data from foreign radars.

2. *The use of measurement techniques that are available too late to be used operationally, or that take measurements with a time step longer than 10 minutes as standard.* In the proposed algorithm for performing reanalyses, in addition to using daily precipitation accumulations provided by those measurement techniques from which data are operationally available, data from manual rain gauges can also be used. These measurements are taken only once a day and are available after about two months – for this reason they are not used in the operational version of RainGRS, but due to their high reliability these data are very important, even crucial.

3. *Greater effectiveness of quality control and data merging algorithms when applied to accumulations longer than 10 minutes, e.g. daily.* Longer precipitation accumulations are more consistent, as they are much less affected by temporal inconsistencies between different measurement techniques (this is especially the case with radar measurements, which in practice are instantaneous), and are moreover less sensitive to errors of a random nature, which become more averaged over a longer time interval. Thus, the algorithms for both quality control and multi-source combination perform more effectively.

At IMGW, combined daily accumulations have been generated since 2021 by the algorithm described in this paper. The resulting daily precipitation estimates can already be directly used to generate longer precipitation accumulations, e.g. monthly, seasonal, annual or even multi-year. In view of the above possibilities, which create new areas of application for multi-source precipitation fields, e.g. in climatology, the version of RainGRS that generates reanalyses of daily precipitation accumulation is referred to as the climatology version RainGRS Clim.

*4.2. Algorithm for the estimation of climatological multi-source precipitation fields*

The algorithm presented in this section for calculating quality-controlled daily and monthly rainfall totals follows the following scheme (Fig. 3):

1. Daily totals are calculated from 10-min rain gauge data. In order to ensure the completeness of the 10-min data, missing rain gauge data is completed with spatially interpolated values from the data that are available. The Ordinary Kriging method is used to interpolate the data.

2. The daily point accumulations from the rain gauges are spatially interpolated to obtain precipitation fields.





3. A human expert check of the daily rain gauge fields is carried out, during which erroneous values from individual rain gauges are removed. This check on the daily values enables the detection of errors that were not detected on the 10-min accumulations with automatic QC algorithms. The daily accumulations from the rain gauges are then spatially interpolated again (as in point 2).

4. Daily accumulations of radar and satellite precipitation fields are calculated, also supplemented with late data.

5. The daily radar precipitation fields are corrected by removing disturbances occurring at the locations of some radars, as this correction only works effectively on longer accumulations.

6. Estimates of daily accumulations $GRS_{reanalysis}$ are calculated by the RainGRS system using the algorithm described in Section 3.1, which uses daily accumulations of individual precipitation fields as input data. This approach minimises errors associated with temporal inconsistencies in the data (Villalobos-Herrera et al., 2022).

7. An adjustment of daily accumulations calculated by the RainGRS to observations from manual rain gauges, which are considered the most reliable point estimate of rainfall, is performed. The adjustment factor is determined separately for each manual rain gauge location and then spatially interpolated using the inverse distance weighting method to distribute it spatially (Wang et al., 2020). This adjustment results in daily accumulations $GRS_{reanal.+adj.}$ of multi-source rainfall fields after reanalysis and adjustment.

8. The long-term accumulations of the combined precipitation fields (e.g., monthly) can be calculated from the daily accumulations prepared in the above manner.



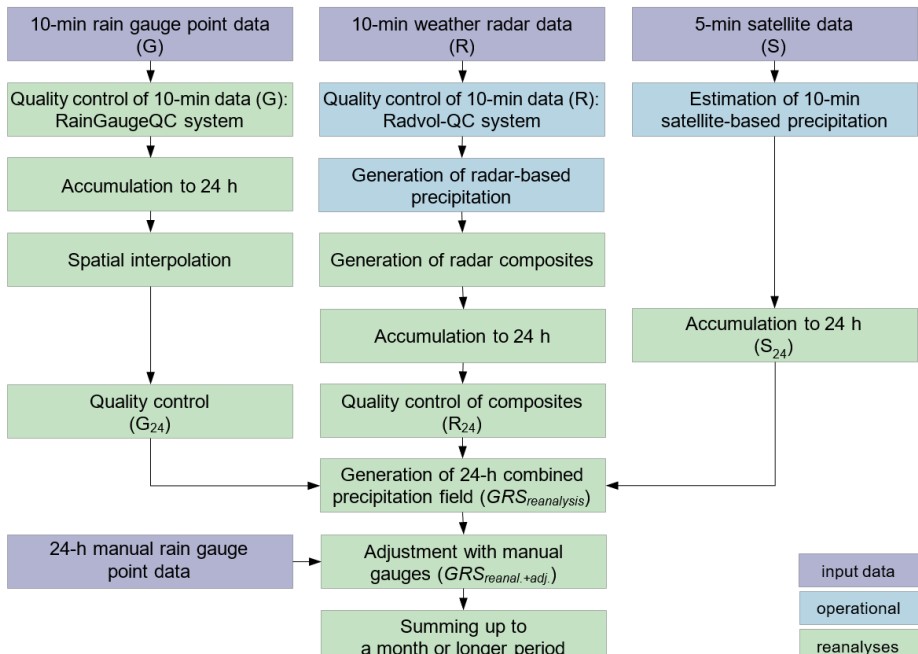

**Figure 3.** The algorithm for determining quality-controlled daily, monthly, and other precipitation accumulations.

Fig. 4 shows an example of daily rainfall accumulations obtained operationally and after reanalysis. The differences between the two fields are generally not large, but locally they can be quite significant – a fragment from the computational domain is selected to highlight them. Larger differences between them are apparent in cases where some rain gauge data have been removed as a result of manual QC (during which they were found to be clearly erroneous) and which was not recognised by operational control. It is likely that in the 10-min accumulations the measurement errors were not so noticeable as to consider these values to be completely erroneous. The removal of each such value also affects the values in a certain vicinity of the rain gauge's location due to changes in the field of interpolated gauges, relevant QI field and consequently in the RainGRS field. In addition, some of the differences between the two fields are due to the varying performance of the data combination algorithm (Sect. 3.1) on daily accumulations when compared to 10-min ones.

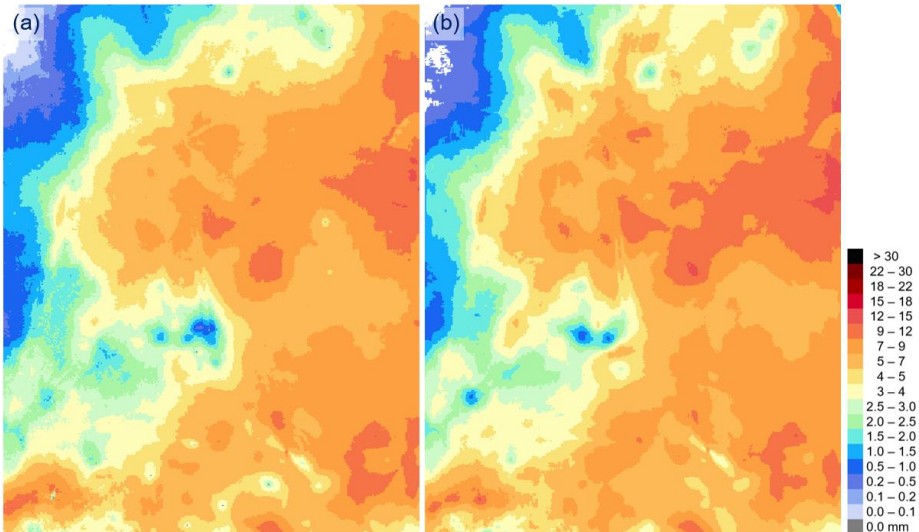

**Figure 4.** Fields of daily precipitation accumulations, before and after reanalysis: (a) $GRS_{real-time}$ and (b) $GRS_{reanalysis}$. Fragment of Poland's computational domain (325 km x 425 km), 11 December 2022.

## 5. Verification

*5.1. Methodology of the verification*

In order to verify any precipitation field estimate, a precipitation field reference that can be considered as a "ground truth" is needed. Lysimeters are regarded as one of the most accurate point precipitation measurement techniques, but Hellmann-type manual rain gauges have similar reliability (Hoffmann et al., 2016). IMGW does not have at its disposal a network of lysimeters, however, it does have a relatively dense network of manual Hellmann type rain gauges, therefore these were considered to provide the most accurate technique of point measurement of precipitation available in IMGW. Thus, the results obtained in the present study were verified on them.

However, it should be borne in mind that the data from the manual rain gauges are not independent, as they have previously been used for adjustment of the RainGRS Clim data. Thus, the basic quantity verified in this Section is not the final precipitation estimates produced after adjustment to the manual rain gauge data, but the estimates after quality control and reanalysis, i.e., $GRS_{reanalysis}$. However, the verification of the final reanalyses $GRS_{reanal.+adj.}$ also provides interesting information, though one should be careful especially with criteria directly related to the estimated values, such as BIAS or RMSE, rather than, for example, their correlation with the reference field.



The period from January 2021 to December 2022 was analysed. For each of these 24 months, the
statistics of the monthly precipitation estimates BIAS, RRSE, RMSE, and CC were calculated, taking
the accumulations from the manual rain gauges as reference:

– statistical bias:
$$\text{BIAS} = \frac{1}{n}\sum_{i=1}^{n}(F_i - O_i) \tag{4}$$

– root mean square error:
$$\text{RMSE} = \sqrt{\frac{1}{n}\sum_{i=1}^{n}(F_i - O_i)^2} \tag{5}$$

– root relative square error:
$$\text{RRSE} = \frac{\sqrt{\sum_{i=1}^{n}(F_i - O_i)^2}}{\sqrt{\sum_{i=1}^{n}(O_i - \overline{O})^2}} \tag{6}$$

– Pearson correlation coefficient:
$$\text{CC} = \frac{\sum_{i=1}^{n}(F_i - \overline{F})(O_i - \overline{O})}{\sqrt{\sum_{i=1}^{n}(O_i - \overline{O})^2 \sum_{i=1}^{n}(F_i - \overline{F})^2}} \tag{7}$$


where $F_i$ is the assessed value, $O_i$ is the reference value (from manual rain gauges), $i$ is the pixel number,
$n$ is the number of pixels, a $\overline{F}$ and $\overline{O}$ are the mean values of $F_i$ and $O_i$.

*5.2. Monthly statistics*

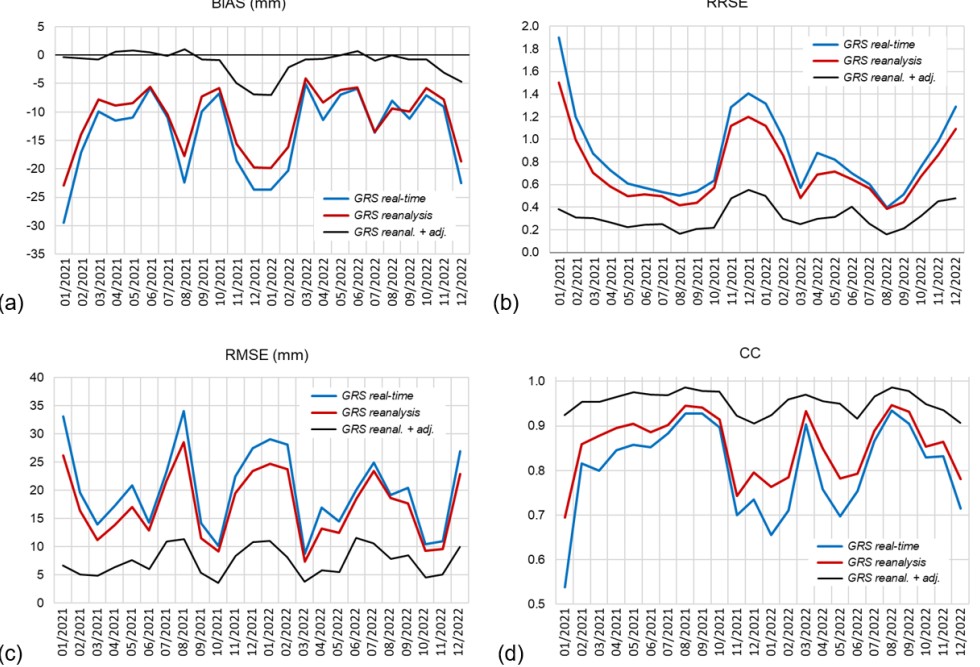

**Figure 5.** Values of monthly characteristics: (a) BIAS, (b) RRSE, (c) RMSE, (d) CC, for precipitation estimates $GRS_{real-time}$, $GRS_{reanalysis}$, and $GRS_{reanal.+adj.}$ for consecutive months, using point data from manual rain gauges as reference. Data for 2021 and 2022.

Figure 5 shows how the values of the four statistics BIAS, RRSE, RMSE, and CC, change in the following months, i.e. depending on the seasonal precipitation characteristics.

The most evident phenomenon visible in the BIAS graph is large underestimation of monthly precipitation accumulations, especially in winter months (December – February) that can reach up to 20 mm (Fig. 5a), which in Poland means several dozen percent of monthly accumulations. This is a result of the fact that the precipitation measurements from both rain gauges and radars are underestimated in IMGW due to the use of specific types of measuring devices, as mentioned in Sections 2.2 and 2.3. Additionally, in winter the reason for these errors is the difficulty in radar measurements that occurs during snowfall from lower clouds than in other seasons and causes most of this precipitation to become invisible to radar as a result of overshooting the precipitation by the radar beam.

Reanalysis and quality control on daily accumulations leads to a reduction of BIAS by a few mm per month, mainly in the winter months. This is mostly due to the clearly better performance of the algorithm for the combination of rain gauge and radar data, which copes better with low precipitation on longer accumulations. After adjustment to observations from manual rain gauges it is possible to deal with the problem of underestimation of the precipitation field – the BIAS is then practically eliminated,





and is visible only to a small extent, mainly in winter. But even then, it is reduced several times, to
approximately -7 mm per month (Fig. 5a). In warmer seasons the observed BIAS values are relatively
smaller, though August 2021 is a clear outlier. Such large errors in this month, visible not only in BIAS
but also in RMSE, are due to the fact that this month was characterised by extremely high precipitation:
the monthly total for a large part of southern Poland was over 300 mm, while in this region the multi-
year average precipitation in August is about 100 mm. High precipitation accumulations are
automatically associated with an increase in the values of statistics of an absolute nature, so that they
are not visible in the values of relative statistics such as RRSE and CC.

The RRSE annual cycle (Fig. 5b) also shows the largest estimation errors in winter. The error is

rather high in winter, at about $1.3 – 1.4$ for $GRS_{real-time}$, and the reanalysis improved the reliability of
the precipitation estimate, resulting in a decrease of the RRSE to a value of about $1.1 – 1.2$. For the other
months, the error is lower, at about 0.5 for $GRS_{real-time}$, and the reanalysis improved the reliability of
the estimate to a lesser extent, as the RRSE decreased by about 0.1.

High values of RMSE (Fig. 5c) are observed in winter, when they reach 27-29 mm for

$GRS_{real-time}$, but unlike RRSE, they also occur in the summer months, which is related to the frequent
occurrence of intense convective precipitation during this season. They do not induce a similar increase
in RRSE values, because this statistic is relative as the result of dividing the RMSE by the standard
deviation from the reference value (Eq. 6). Reanalysis reduces RMSE values in winter by about 5 mm
per month, slightly less in the other seasons, and adjustment to manual rain gauges reduces them to
values of about 5-10 mm per month independently of the season.

The correlation coefficient CC (Fig. 5d) is more sensitive to the existence of relationship between

evaluated and reference data than the other statistics, which are based on the comparison of estimated
and reference values. The CC values also indicate the lowest reliability of the precipitation estimates in
winter, when the coefficient equals about 0.65 and improves to about 0.75 after reanalysis. The reason
for these low values can also be explained by the low variability of the precipitation accumulations over
this period, which results in a low correlation with the manual rain gauge measurements. In other
seasons, especially in the summer months, the CC values are much higher, as they reach approximately
$0.8 – 0.9$ for both operational and reanalysed estimates. The adjustment to the manual rain gauges
increases the correlations to approximately $0.9 – 0.95$.

In March 2022, there was a noticeable deviation from the typical annual pattern described above

for the CC coefficient. This was due to the exceptionally dry period that occurred at that time in the
whole country, particularly in northern Poland. Typically, monthly precipitation accumulation for
March is around 30-40 mm in Poland, but in 2022 it was significantly lower, and in the northern part of
the country it was often even zero. In this case, the correlation coefficient usually increases, so that in
this particular month, the correlation value for $GRS_{real-time}$ was as high as 0.90, rising to 0.93 after
reanalysis. Another unexpected value of the CC coefficient was observed in May 2022, when the
correlation is around 0.7, which was improved by reanalysis and adjustment, after which the CC





increased to around 0.95. The reason for this effect was probably a Legionowo radar replacement at that
time, because this radar covers a large part of the domain where other radars do not reach.
In general, the reliability of monthly estimates of precipitation field accumulation is clearly
dependent on the season. Two evident phenomena can be observed here: in winter (November –
February), high values of BIAS, RRSE, and RMSE are noticeable at the same time as low values of CC,
as indicated in the above analysis. In summer (July – August), the situation is different, as convective,
thunderstorm precipitation is often observed during this time, so the intensity of precipitation is higher,
and monthly accumulations are much higher, which is also reflected in the RMSE values, while the
correlation (CC) with the reference data is then significantly higher.
Table 2 summarises statistics for two selected months from 2022: January for winter and August
for summer. The table shows the values of quality metrics for the three multi-source precipitation fields:
operationally generated ($GRS_{real-time}$), after reanalysis ($GRS_{reanalysis}$), and after adjustment of this
reanalysed precipitation field ($GRS_{reanal.+adj.}$), with manual rain gauge observations as a reference. All
statistics are worse for winter than for summer, however, reanalysis as well as adjustment worked much
more effectively in winter. Precipitation reanalysis, involving merging individual rainfall fields on daily
(instead of 10-min) accumulations, along with the associated more effective data quality control, results
in a clear improvement in all quality statistics in winter (January 2022), e.g. RMSE by almost 4.5 mm
and CC by 0.1. In summer (August 2022), however, this impact is much smaller, and amounts to less
than 0.6 mm and 0.02, respectively, but BIAS slightly increased. The further improvement, which results
from adjustment to data from manual rain gauges, is much more evident – in winter it is more than 13.5
mm in RMSE and 0.16 in CC, and in summer more than 11.8 mm and 0.04, respectively.

Table 2. Values of quality metrics for merged daily precipitation fields: before reanalysis ($GRS_{real-time}$), after
reanalysis ($GRS_{reanalysis}$), and after reanalysis and adjustment ($GRS_{reanal.+adj.}$), using point data from manual
rain gauges as reference. Months: (a) January 2022, (b) August 2022.

(a) January 2022

| Metric | BIAS (mm) | RMSE (mm) | RRSE (--) | CC (--) |
|---|---|---|---|---|
| $GRS_{real-time}$ | -23.72 | 29.04 | 1.32 | 0.66 |
| $GRS_{reanalysis}$ | -19.83 | 24.63 | 1.12 | 0.76 |
| $GRS_{reanal.+adj.}$ | -7.06 | 11.06 | 0.50 | 0.92 |


(b) August 2022

| Metric | BIAS (mm) | RMSE (mm) | RRSE (--) | CC (--) |
|---|---|---|---|---|
| $GRS_{real-time}$ | -8.04 | 19.18 | 0.40 | 0.93 |
| $GRS_{reanalysis}$ | -9.35 | 18.60 | 0.38 | 0.95 |





| $GRS_{reanal.+adj.}$ | -0.03 | 7.77 | 0.16 | 0.99 |
|---|---|---|---|---|


Concluding, for all the statistics used here, the improvement in the quality of monthly
accumulation of estimated precipitation fields $GRS_{reanalysis}$ i $GRS_{reanal.+adj.}$ relative to operational
fields $GRS_{real-time}$ is clearly visible. The differences between the statistics of $GRS_{reanal.+adj.}$ and
$GRS_{real-time}$ are much larger. This is mainly due to the fact that, in the absence of any other possibility,
the verification was carried out using data from manual rain gauges as a reference, and here they are
dependent data, as they are used during the generation of the final $GRS_{reanal.+adj.}$ (see point 7 in the
data processing scheme in Section 4.2).

(a) January 2022

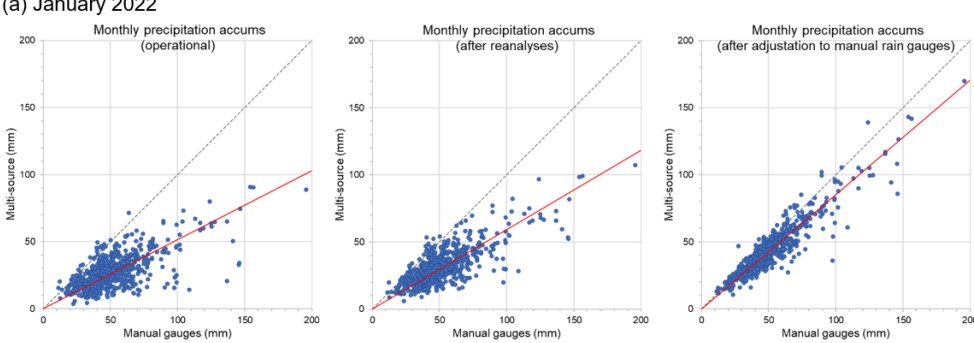

(b) August 2022

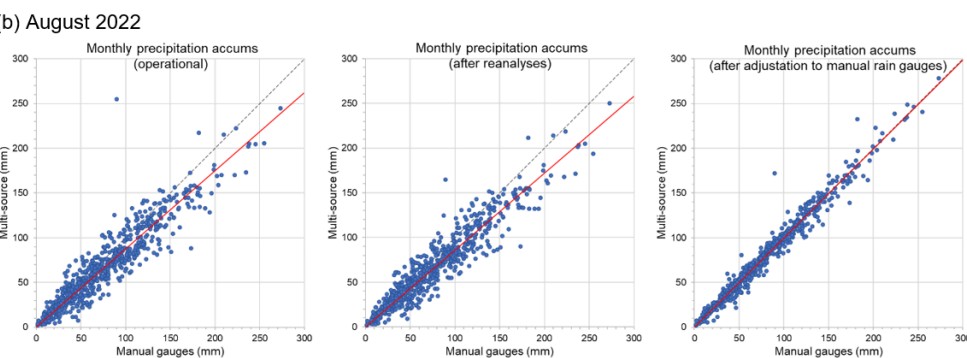


**Figure 6.** Plots of the dependence of monthly precipitation estimate values, from left: $GRS_{real-time}$, $GRS_{reanalysis}$
and $GRS_{reanal.+adj.}$ on values measured with manual rain gauges, along with trend lines. Months: (a) January
2022, (b) August 2022.

Fig. 6 shows graphs of the relationship between the estimated fields of monthly accumulated
RainGRS precipitation calculated operationally (generated in real-time), after reanalysis and after
adjustment of this reanalysed precipitation field, and monthly accumulations observed by manual rain





gauges, for the same two months for which the values of statistics are summarised in Table 2. The graphs
show precipitation values at locations of manual rain gauges. The correlation for the $GRS_{reanalysis}$
estimate compared to $GRS_{real-time}$ improved, although only slightly. This conformity, measured by the
distance between the trend line (red) and the one-to-one line (dashed), clearly improved in winter, but
declined slightly in summer. The conformity with manual rain gauges for the $GRS_{reanal.+adj.}$ estimate
is clearly greater than that for the $GRS_{reanalysis}$, but it should be borne in mind that the data from manual
rain gauges are not fully independent. Nevertheless, this comparison gives some information about the
effectiveness of the final step in generating precipitation field estimates with the RainGRS Clim system.

**6. Example of a climatological estimate of monthly precipitation accumulation**

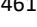
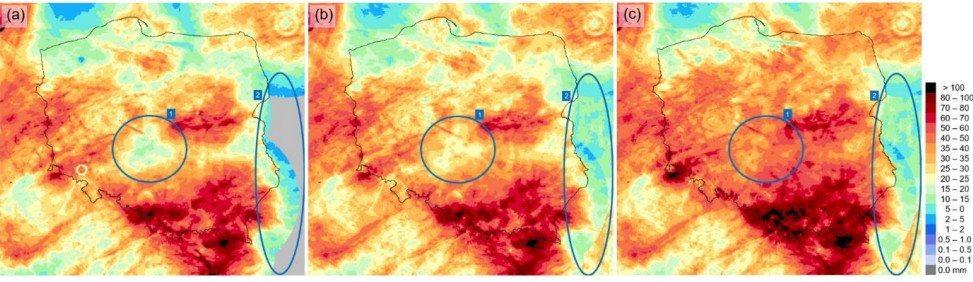



**Figure 7.** Fields of monthly precipitation accumulations: (a) $GRS_{real-time}$, (b) $GRS_{reanalysis}$, and (c)
$GRS_{reanal.+adj.}$. Domain of Poland, April 2021.

In Fig. 7 we can see an example of estimates of monthly precipitation accumulations for the
domain of Poland, 900 km x 800 km (see Fig. 2). From the left there are estimates: operational, after the
reanalysis, and after reanalysis and adjustment to manual rain gauges data. In general, values of the
estimated precipitation increased after the reanalysis as a result of the more effective performance of the
merging algorithm on longer accumulations. After the adjustment to manual rain gauges, the further,
much higher increase of the precipitation values is because radar-based precipitation estimates are
underestimated in the case of Polish weather radars. Moreover, it should be taken into account that rain
gauges also underestimate rainfall, because they are mostly tipping bucket devices (Segovia-Cardozo et
al., 2021).
The area of underestimated precipitation in the centre of Poland marked with "1" in Fig. 7 is the
place where the distance to the closest radar site is longest – more than 200 km, where the radar beam
passes over part of the precipitation (overshoots). Moreover, the telemetric rain gauge network is rather
sparse here. Adjustment to manual rain gauges has made it possible to correct this underestimation.
The area denoted "2" in Fig. 7 indicates the region where there are no radars, even from
neighbouring countries. Reanalysis partially improves it by complementing the lack of data with



satellite-based precipitation, but not wholly effectively due to the higher uncertainty of the satellite
estimates.

**7. Conclusions**

The following general conclusions can be drawn about the proposed methodology for the
generation of long-term precipitation estimates by the RainGRS Clim system:

1.  Based on an analysis of available precipitation data, it was assumed that the most reliable
precipitation measurement technique is a network of manual rain gauges. In particular, it was
assumed that these measurements are unbiased. Since their daily accumulations are available
with a long delay due to their transfer and manual quality control, they cannot be used in real
time, but they can be used effectively to perform adjustment of reanalyses (see Sections 5.2
and 5.3).

2.  The second major limitation of manual rain gauges is that they only provide point observations.
However, the relatively high density of this measurement network in Poland (Fig. 1) makes
them very useful in the adjustment of other precipitation field estimates.

3.  With daily accumulations, which, due to the time step of manual rain gauge measurements,
are the basic accumulations in the algorithm for generating climatological precipitation
estimates described in Section 4.2, it becomes possible to perform much more effective quality
control, particularly in terms of removing various types of artifacts in weather radar data.

4.  Algorithms for merging rain gauge, weather radar, and satellite data perform much more
effectively for daily totals than for 10-min totals. This is mainly due to the fact that longer
accumulations of precipitation are more consistent, as in this case time inconsistencies
between different measurement techniques play a much smaller role. In addition, with longer
accumulations, errors of a random nature are more averaged out (see Section 4.1).

5.  The results presented in the paper show that after reanalysis, estimates of precipitation field
are of higher reliability than operationally generated estimates. Adjustment of the data after
reanalysis to data from manual rain gauges resulted in a further, much higher quality
improvement (Sections 5.2 and 5.3). However, it should be kept in mind that the final estimates
are obtained using data from manual rain gauges, so the results of the verification performed
on these data, which in this case are partially dependent, should be treated with caution.

6.  Having estimates of precipitation accumulated over longer time intervals in RainGRS Clim,
such as monthly intervals, creates the possibility of applying them to climatological analyses.
They provide valuable information, especially when high spatial resolution of precipitation
data is important.






*Code availability*. The data processing codes are protected through the economic property rights to the software
and are not available for distribution. The codes used for processing follow the methodologies and equations
described herein.

*Data availability*. The data used in this paper are available upon request.

*Author contributions*. AJ, KO, JS, and MP designed algorithms of the RainGRS Clim system. MP, KO, and AK
developed the software code and performed the simulations. JS, KO, AJ, AK, and MP prepared the paper. JS made
figures.

*Competing interests*. The contact author has declared that none of the authors has any competing interests.

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
