# Peer review of "1. Introduction"

_Atmospheric Measurement Techniques, 2023_

## Author Response (AR1)

**RC1**: ['Comment on amt-2023-98'](), Anonymous Referee #1, 22 Jun 2023

An interesting and well composed article.

Just a minor technical detail: by "manual rain gauges" (lines 330, 341) do you mean your Hellmann-type gauges? The type could be specified here for clarity.

An idea to potential further studies. You mention beam overshooting in the case of low-initiating winter precipitation. Could such conditions be recognized dynamically, and used as/in a quality index in the proposed observation combining scheme?

**Answer**

Thanks to the reviewer for the opinion and comments!

The information that the manual rain gauges used in our work are of the Hellmann type is very important, of course. Such specification can be found in the paper on line 130, in section 2.2 "Rain gauge data".

Your comment about the possibility of dynamic overshooting recognition in case of winter precipitation is very interesting. It is possible in the case of partial overshooting to roughly estimate what part of the precipitation has been overshooted, as long as it is not convective. But what to do in the case of total overshooting, i.e. at greater distances from the radar? We could lower the QI a priori at these locations on the basis of some statistical relationship, but then it would not be a dynamic but a static quality factor related only to the distance from the radar, and we already have this kind of factors in the applied QI scheme. However, we will consider what the possibilities are for a dynamic approach to recognising this phenomenon because it is, as it turns out, important for the estimation of the precipitation field.

**RC2**: ['Comment on amt-2023-98'](), Anonymous Referee #2, 02 Jul 2023

The manuscript is well written (just singular misprints) and structured. The methodology is clear and well described. It covers the important issue of providing the complex information about precipitation field in climatological resolution.

It seems to be promising and powerful tool for further analyses.

**Answer**

Many thanks to the reviewer for the positive assessment of our paper! Of course, we will read our text carefully in terms of searching for all misprints.